# Ambient Noise in Candidate Rooms for User-Operated Audiometry

**DOI:** 10.3390/healthcare11060889

**Published:** 2023-03-20

**Authors:** Christos Sidiras, Jacob Nielsen, Chris Bang Sørensen, Jesper Hvass Schmidt, René Gyldenlund Pedersen, Ellen Raben Pedersen

**Affiliations:** 1Clinical Psychoacoustics Lab, 3rd Psychiatry Department, School of Medicine, Faculty of Health Sciences, Aristotle University of Thessaloniki, 54636 Thessaloniki, Greece; 2The Maersk Mc-Kinney Moller Institute, University of Southern Denmark, 5230 Odense, Denmark; 3Department of Clinical Research, Faculty of Health Science, University of Southern Denmark, 5230 Odense, Denmark; 4Research Unit for ORL—Head & Neck Surgery and Audiology, Odense University Hospital, 5000 Odense, Denmark; 5OPEN, Odense Patient Data Explorative Network, Odense University Hospital, 5000 Odense, Denmark

**Keywords:** user-operated, pure tone audiometry, ambient noise, sound treated, acoustic environment

## Abstract

Hearing loss is a widespread problem while treatment is not always accessible, mainly because of the limited availability of hearing care professionals and clinics. In this work, part of the User-Operated Audiometry project, we investigate the acoustic environment of inexpensive non-sound-treated rooms that could be used for unsupervised audiometric testing. Measurements of 10 min of ambient noise were taken from 20 non-sound-treated rooms in libraries and private and public clinics, nine of which were measured twice. Ambient noise was compared against two traditional audiometric sound-treated rooms and Maximum Permissible Ambient noise levels by ISO 8231-1, while factoring for the attenuation by the DD450 circumaural headphones provided. In most non-sound-treated rooms, MPAs were violated only by transient sounds, while the floor-noise level was below MPAs. Non-sound-treated rooms’ ambient noise levels presented with much larger fluctuations compared to sound-treated rooms. Almost all violations occurred at low to mid-low frequencies. Our results suggest that large-scale implementation of user-operated audiometry outside traditional audiometric rooms is possible, at least under some realizable conditions. Circumaural headphones’ attenuation is probably a necessary condition for all cases. Depending on the room, an online system making decisions based on ambient noise might also be included in combination with active attenuation.

## 1. Introduction

Hearing loss is a widespread problem especially for older adults. It impacts social and professional life, cognitive function, psychological health, and economy at both the personal and the societal level [1,2,3,4,5]. Treatment is accessible to only some patients, mainly because of the limited availability of hearing care professionals and clinics [6,7]. Efforts are being made towards the development of novel audiometric testing that fills this gap, including the User-Operated Audiometry (UAud) project [7,8,9]. The UAud solution includes direct unsupervised interaction of patients with an automated system located at easily accessible testing rooms. In a recent work, we addressed one of the challenges faced by the UAud project by investigating whether patients are capable of placing headphones themselves adequately [10]. In this work, also part of the UAud project, we focus on the acoustic environment in rooms that can be used for the UAud solution.

Ambient noise affects audiometric testing, either by masking or by causing distractions [11]. In this regard, specific standards on maximum limits of ambient noise have been developed to ensure minimal to no effects of ambient noise on the obtained hearing thresholds, such as ANSI S3.1-1999 and ISO 8231-1 [12,13,14]. Within the traditional audiometry paradigm, complying to these standards is achieved by using expensive sound-treated rooms that require special installation, though in practice these standards are rarely met in a strict sense [15,16]. This poses serious limitations on the number of available testing locations, which contributes to the hearing health accessibility issue.

Efforts towards a solution to the limited accessibility issue have been made by exploring testing outside sound-treated rooms. There are quite a few studies comparing audiometric results while testing in sound-treated rooms vs. non-sound-treated ones. These studies include various combinations of passive and active headphone attenuation, the use of earmuffs, insert earphones, and ambient noise monitoring, while testing room locations include, among others, schools and retirement facilities [9,17,18,19,20,21,22]. The results so far are encouraging, though they may be not enough to change the established sound-treated-room paradigm in audiometry.

The solution under investigation here includes the use of a specific circumaural headphone set, the DD450 audiometry headphones, and venues with specific characteristics that take into consideration the main target group of UAud which is older adults. The consideration are as follows: first, UAud candidate rooms must be easily accessible via public transportation and by car. Second, they must offer a comfortable environment, free from visual or other distractions [11]. Third, a clerk must be available during testing hours for any test-related needs. Last, ambient noise during testing hours must not affect the testing.

This study is a first step in addressing the potential effects of ambient noise in candidate rooms for the UAud solution. Specifically, the purpose is to describe ambient noise in detail; this description ought to be relevant to the characteristics of the audiometric testing procedure and to the ambient noise golden standards in audiology, making patterns in ambient noise visible, and allowing for future investigation of the effects of ambient noise on audiometry. The study method includes 10 min recordings, which is close to the duration of a pure-tone hearing test, and compares them against the Maximum Permissible Ambient noise levels (MPA) set by ISO and ambient noise in sound-treated rooms used in traditional audiometry while factoring in the attenuation of the headphones used in the UAud project.

## 2. Methods

### 2.1. Rooms

Ambient noise was measured in 20 UAud candidate rooms. These were all non-sound-treated rooms, nine of which were measured twice for test–retest comparison (see Table 1 for details on demographic data). All rooms had one or two external walls, hence sound from outside might contribute to ambient noise inside the room. Thirteen rooms were located at the audiology clinic, Odense University Hospital (OUH). Two rooms were at a private ENT clinic in Odense and one room was at the Southern University of Denmark (SDU) teaching clinic. These 16 rooms were relatively small, around 5–10 m^2^. Four larger rooms, either above 50 m^2^, or big foyer-like rooms were also measured. Two rooms were at the SDU Odense library and two at the Odense public library at Citizen’s House. Additionally, measurements were collected from two sound-treated rooms used for standard manual audiometric testing located in the audiology clinic (OUH) for comparison.

### 2.2. Measurements

A 6th generation iPad (Apple), model number A1893, was used for measuring background noise. The software used was the AudioTools app (Studio Six Digital) with the additional SPL Graph module (For SPL Pro software specifics see https://studiosixdigital.com/audiotools-modules-2/spl-modules/spl/) (newest release as of 2 December 2021). The iPad was calibrated by adjusting the sensitivity of the iPad microphone according to the measured sound level of white noise of a B&K 2236 sound-level meter. The calibration was performed in an acoustically well-regulated office room, where a homogenous sound field in the measuring area can be created with no difference in the sound pressure at the two microphone positions (B&K and iPad). The iPad was placed where a patient would be seated, and the built-in microphone directed towards the door, for most rooms, or towards the direction of the loudest source of noise according to experimenter’s judgement. The measurements took place during rush hours/days specific for each room according to the local personnel, avoiding holidays and COVID lockdowns. Test and retest measurements took place on different dates to investigate for fluctuations of ambient noise across days.

Each measurement lasted 10 min. While most measurements (*n* = 24) ran smoothly, during some (*n* = 8) measurements, unexpected noises were produced when someone from the personnel opened the door of the room. These short segments were cleared out from our data set as they are not part of the expected ambient noise during UAud testing. The software was set to extract dB_SPL_ levels of 25 one-third octave frequency bands (center frequencies: 32–8000 Hz), one sample per second. In total, the data set fed into subsequent analysis consisted of 31 measurement sets (20 non-sound-treated rooms tested, plus retests for nine of them, plus two sound-treated rooms), times 25 frequency bands, times ~600 samples (one per second for 10 min and accounting for the cleared parts), which corresponds to a total of ~465.000 data points. The 9 rooms that were retested were chosen randomly out of the total 13 OUH rooms.

### 2.3. Data Analysis

Data analysis focused on how dB_SPL_ compared to Max Permissible Ambient Sound Pressure Level (MPA_SPL_; [13]), and thus derived measures were extracted from data points. dB_MPA_ is the standardized sound level accounting for the MPA_SPL_. It is equal to dB_SPL_-MPA_SPL_. A dB_MPA_ value is equal to 0 when a dB_SPL_ value is exactly equal to the MPA_SPL_, while positive and negative values are higher and lower, respectively. Two additional measures were extracted only for violations. Violations are data points that lie higher than the Max Ambient Permissible Sound Pressure Level (MAP_SPL_). The additional measures were the total number of violations in each 10 min measurement (V_n_10) and the magnitude of each violation, V_mag_, equal to dB_SPL_-MPA_SPL_. The difference between dB_MPA_ and V_mag_ is that the former applies for all data points and can be either positive or negative, while the latter applies only to violations and is strictly positive.

MPA_SPL_ values were set according to Table 2 of ISO 8253-1 [13], which applies to air conduction audiometry at frequencies from 250 Hz to 8000 Hz and differed according to the room measured. For non-sound-treated rooms the attenuation that the DD450 circumaural headphones provided was taken into account. This was done by subtracting the typical supra-aural headphones’ attenuation that is factored in Table 2 of ISO 8253-1, and then adding the attenuation provided by DD450. Figure 1 shows the respective MPA_SPL_ values when DD450 and typical supra-aural headphones were used, and the attenuation they provide. For the two sound-treated rooms used for traditional manual audiometry, the original MPA_SPL_ values were used instead. Note that MPA_SPL_ settings affected all derived measures (dB_MPA_, V_n_10 and V_mag_).

Analysis and figures were executed and produced in MATLAB version R2021b. Statistical analysis included non-parametric descriptive statistics, i.e., min, Q1, median, Q4, and max.

## 3. Results

### 3.1. UAud-Candidate Rooms (Non-Sound Treated Rooms)

Visual inspection of the time vs. dB_MPA_ plots of each frequency band was executed for each measurement. This revealed that violations fell into two categories: quasi-constant violations vs. brief ones. Quasi-constant violations were due to quasi-steady noise, while brief ones were due to transient sounds (e.g., footsteps). Figure 2A,B show ambient noise of the 200 Hz band in a room with quasi-constant violations and a room without quasi-constant violations, respectively. Two things should be noted; first, quasi-steady noise did violate ISO MPA_SPL_ values for typical supra-aural headphones, instead of circumaural ones (DD450; see Figure 2B). This highlights the importance of the attenuation that circumaural headphones provide, as was the case for all rooms tested. Management strategies and potential effects on audiometric testing may differ significantly between constant vs. non-constant noise violations. Thus, the two categories should be considered separately.

Rooms 1 to 16, that is, the audiological clinic at OUH, the private clinic, and SDU examination rooms, did not contain quasi-constant violations (Table 1). Rooms 17 to 20 (*n* = 4), that is, the two rooms in SDU library and the two rooms at Odense public library, contained either both brief and quasi-constant violations, or quasi-constant violations only. These two groups were fed into subsequent analysis separately.

#### 3.1.1. Rooms without Quasi-Constant Violations

Figure 3A displays the descriptive statistics of V_n_10 in boxplots and V_mag_ in lines and shaded areas for all rooms without quasi-constant violations combined (rooms 1 to 16). Almost all violations occurred in frequency bands from 160 to 630 Hz. The 200 Hz band had the highest median V_n_10 (equal to four), and the distribution was positively skewed, reaching a maximum of 34 violations. The next most frequently violated frequency bands were the 250 Hz, 315 Hz, and 400 Hz bands, with V_n_10 medians equal to two, two, and one, respectively. In particular, the V_n_10 distribution of the 250 Hz band was particularly skewed reaching a higher maximum than the 200 Hz band. This was due to the contribution of two single cases, room 6 and 15 (also see Figure 3B). Median V_mag_ was uniform and lower than 5 dB_SPL_ across the whole frequency spectrum (see solid grey line in Figure 3A). Skewness towards larger values was also observed, reaching up to a max of 28 dB for the 200 Hz band (see dashed line in Figure 3A).

For most rooms, the most frequently violated frequency bands were 200 Hz, 250 Hz, and 315 Hz (see Table 2). Figure 3B displays the V_mag_ non-parametric descriptive statistics vs. V_n_10 of each room’s most frequently violated frequency band. For half of the rooms (*n* = 6) violations were less than ten with a magnitude less than 10 dB. For two rooms, V_n_10 was particularly high (rooms 6 and 15; V_n_10 equals 86 and 34, respectively), but V_mag_ values were rather small (<8 dB).

#### 3.1.2. Rooms with Quasi-Constant Violations

Figure 4 displays dB_MPA_ non-parametric descriptive statistics per frequency band when circumaural headphones (DD450) were used for rooms with quasi-constant violations (rooms 17 to 20). Median ambient noise violated MPA_SPL_ up to 5 dB within frequency bands from 200 Hz to 500 Hz in all rooms except room 18. A large portion of ambient noise reaching up to 25 dB_MPA_ was seen in higher frequency bands as well, up to 2000 Hz. No violations were seen for frequency bands above 3100 Hz in any room, except for room 19, where a peak at 8000 Hz was present. This indicates the potential presence of noise at higher frequencies for which there is no data.

#### 3.1.3. Test–Retest

Retest recordings were run for nine rooms (1, 2, 4–7 and 10–12). All of them were rooms in which quasi-constant violations were absent during the first test (see Table 1). Visual inspection of time vs. dB_SPL_ plots for each frequency band revealed that quasi-constant violations were absent in the retests as well.

Figure 5A and 5B display test and retest descriptive statistics of V_n_10 in boxplots and V_mag_ in lines and shaded areas for the retested rooms (note that Figure 4A differs from Figure 2A, as the former refers only to rooms that were rested (*n* = 9), while the latter to all rooms without quasi-constant violations (*n* = 16)). For both test and retest, almost all violations occurred at frequency bands from 200 Hz to 400 Hz. There was some difference in V_n_10 between the test and retest, with retest medians being higher than test ones by up to 3 violations for the 250 Hz band. The V_mag_ median for both test and retest did not exceed 5 dB for the whole spectrum, while max values were up to 10 dB lower in the retest.

Table 3 shows test–retest V_n_10 and V_mag_ medians and max as well as their differences for each room’s most violated frequency band, either in the test or retest. For some rooms (5, 11 and 12), V_n_10 did not differ by much between the test and retest (difference <4) while for others (4, 6 and 10), large differences (>74) were observed. Differences in V_mag_ medians were lower than 3 dB except for room 7 (equal to 5.5 dB), and differences in V_mag_ max were around 5 dB or less, except for rooms 1 and 2 (equal to 20 dB and 15 dB, respectively).

### 3.2. Standard Audiometry Rooms (Sound-Treated Rooms)

Figure 6 displays the non-parametric descriptive statistics of dB_MPA_ per frequency band for the two sound-treated rooms. Note that the MPA_SPL_ used for these rooms refer to use of typical supra-aural headphones. Ambient noise is much lower compared to non-sound-treated rooms for which MPA_SPL_ refers to the use of DD450 circumaural headphones.

The ambient noise was similar between the two rooms and were mostly below MPA_SPL_. Violations were observed at the frequency bands from 200 Hz to 800 Hz. For the first room, ambient noise peaked at the 315 Hz band and was higher than MPA_SPL_ throughout the whole 10 min recording (dB_MPA_ median = 1.9 dB and max = 4.6 dB). For the second room, dB_MPA_ median was below zero across the whole spectrum, and peaked at 160 Hz and 800 Hz, with 8.5 dB and 5.1 dB, respectively.

## 4. Discussion

### 4.1. MPA Violations—Overview

The main findings of this study are the following. First, sound-treated rooms’ ambient noise does violate MPAs, though these violations are rather small in magnitude. Note that in one room, ambient noise violated MPA for the 315 Hz band during the whole 10 min measurement’s duration. Second, in most non-sound-treated rooms, that is all rooms in clinics, MPAs were violated only by transient sounds, while the floor-noise level was below MPAs. This study was not designed to investigate the sound sources, though some speculations can be made. As clinics are quite busy and people were walking though the corridor, it is most probable that most sounds were people speaking or sounds related to movement, e.g., steps. Some sounds might also have come from outside, such as traffic. Third, non-sound-treated rooms’ ambient noise levels present with much larger fluctuations compared to sound-treated rooms. Last, almost all violations, in both sound-treated and non-sound-treated rooms, occurred at low to mid-low frequencies. Given that small violations seen in sound-treated rooms seem to not pose a serious problem in traditional audiometry [15,16], it seems that audiometry in at least some non-sound-treated rooms is possible. It should be noted though that the large noise fluctuations over time observed in the latter rooms should be taken into consideration.

In the standard audiometric sound-treated rooms, ambient noise median levels were mostly below the ISO MPA values. Due to fluctuations in time, violations reaching up to 8 dB were recorded within the mid-low frequency range. Frank and Williams [15] measured noise levels from 136 audiometric testing rooms and found that half of them violated the 250 Hz band by 3 dB or more, according to the standards used here. Similarly, Kim et al. [16] measured 124 audiometric test rooms and found that only 13.7% had no violations, though they used ANSI standards which are a bit stricter, around 5 dB lower compared to ISO. These are the only studies measuring ambient noise in rooms used for clinical audiometry that the authors are aware of. Based on these studies, it can be inferred that the rooms tested in the present study lie on the quiet side of the spectrum when compared to the average. It should be noted though, that this comparison has some limitations because of the lack of details about the duration of the recordings and how the reported levels were computed, and the fact that no between octave frequency bands were measured.

The ambient noise spectrum in library non-sound-treated rooms was similar to sound-treated rooms. They peaked at low to middle frequencies and progressively decreased at higher frequencies. Median levels in three out of four rooms did not violate MPAs more than 5 dB, while in one room they were entirely below MPA. However, the ambient noise level’s range was much larger compared to sound-treated rooms, being 20–30 dB vs. ~10 dB, respectively, resulting in violations that reached up to almost 30 dB. Note that headphone attenuation was taken into account for the respective MPA values.

Rooms in clinics presented a very different profile compared to library rooms; constant noise was below MPAs, and only brief violations were observed. This difference, along with the lack of studies investigating brief violations, makes comparison with other ambient noise profiles challenging. In more than half of the clinic rooms, violations were few and low in magnitude (around 10 dB). Still, some rooms were quite noisy, in which ambient noise often exceeded MPAs by up to 20–30 dB. Across all clinic rooms, frequency bands that violated MPAs were essentially the same as in the library rooms.

The large difference in ambient noise level fluctuations between sound-treated vs. non-sound-treated rooms, both in clinics and in libraries, may be caused by differences in the nature and position of sound sources. Sound-treatment attenuates a big part of the sound from human activities coming from outside, such as speech and steps. The remaining ambient noise is coming either through the walls or from equipment within the room, such as computers, ventilation, or lights, which is more or less constant through time. In non-sound-treated rooms, the sound from human activities coming from outside or the rest of the area for the very large library spaces, is the one that dominates. The larger range in these rooms seems to be the direct result of large fluctuations of human activity-related sounds.

### 4.2. Test–Retest—Limitations and Generalizability of Results

Test–retest measures applied in nine rooms with only transient violations revealed some differences in ambient noise at different times. In three out of the nine rooms that were remeasured, the number of violations differed by more than 15. Regarding the violations’ magnitude though, there was only one room in which the mean difference exceeded 5 dB. Frequency bands in which violations occurred were the same, that is mid-low frequencies, as was the type of the violations, that is, only due to transient sounds. It should be noted that as no room with quasi-constant violation was retested, this result may have some limitations.

Test–retest differences might be a limitation in the methodology, as our results suggest that a one-time 10 min recording lacks information about ambient noise differences for different times and days. Further, 10 min recordings might miss sound events such as train traffic near train stations. An alternative interpretation though could be that this result shows which changes are to be expected in ambient noise and which are not. Frequency and magnitude of violations are expected to stay more or less the same at different times, as with presence or absence of quasi-constant violations. On the other hand, the number of violations is expected to fluctuate over time. It should be noted that these results apply to the specific rooms tested in this study and generalization over other possible venues might not be possible.

### 4.3. Challenges in Predicting Ambient Noise Effects

ISO 8253-1 [13] states that when MPAs are not violated, an uncertainty of 2 dB is expected for a lowest threshold level of 0 dB, and 5 dB if ambient noise is 8 dB above MPAs. Typically, hearing loss is treated with hearing aids if thresholds reach 30 dB HL and above. In some situations, treatment might be needed in the 20–30 dB HL interval. Thus, accuracy of hearing thresholds is of great importance as they inform medical decisions. Further, it seems reasonable to hypothesize that the less ambient noise is perceived, the less its effect on audiometry would be. This would entail that ambient noise should be less of a concern in large hearing loss. To the authors’ knowledge, no study exists looking into the relationship between hearing loss and ambient noise effects. Perhaps the biggest challenge is predicting effects when ambient noise levels fluctuate over time, especially for transient sounds which were the most frequent case in the present study.

Wong et al. [23] measured absolute differences in thresholds for manual audiometry with supra-aural headphones in sound-proof vs. several non-sound-proof rooms. Average ambient noise in the latter ones was around 45 dB at 500 Hz and progressively decreasing for higher frequencies, but varied a lot among rooms (SDs > 10 dB). They found larger differences at low frequency thresholds which progressively decreased towards higher frequency thresholds (~9 dB at 500 Hz, <2 dB at 8000 Hz). In a similar study, Swanepoel et al. (2013) used insert earphones and circumaural earcups on-top and found no systematic differences between sound-treated rooms vs. rooms with an average of 55 dBA of ambient noise. A limitation in these and other similar studies, is the lack of time domain details [15,24]. Differences in the methods used to measure ambient noise makes comparisons difficult.

Apart from masking, potential effects of ambient noise on cognition should also be considered. Ambient noise can function as a distractor, or cause irritation and fatigue [11]. This probably applies to sounds occurring not only during stimuli presentation, but during the whole testing. The effects might be pronounced for the case of transient sounds and sounds that carry semantic information, such as footsteps or speech. Further, effects on cognition, if present, would probably not be uniform across patients, but rather depend to a degree on individual cognitive skills, such as attention. Here, is a short summarized list of questions for future research:Does ambient noise effects depend on hearing loss?What is the effect of transient violations in audiometry?Are there any effects other than masking and is ambient noise’s semantic content relevant?Do patients’ characteristics, such as distractibility, discomfort to sound, and/or cognition play a role?

The authors think that bridging these gaps in knowledge would be a step forward for implementing user-operated audiometry in multiple easily accessible and inexpensive sites. In addition, this knowledge will be useful in informing a system when to pause testing when needed (see Section 4.5 ‘Addressing Ambient Noise—Online Test Pause’).

### 4.4. Ambient Noise Violations and Headphones Attenuation

A common pattern observed across rooms is that the likelihood of ambient noise violating MPAs is not uniform across the frequency spectrum. The frequencies at which all violations were observed at were within the 200–800 Hz range, peaking around 250 Hz, and decreasing towards higher frequencies. This pattern was present quite consistently across all rooms in this study, sound-treated and non-sound-treated. This pattern has also been observed in sound-treated rooms in other studies as well (Frank & Williams, 1993; Kim et al., 2004). Our results suggest that the DD450 headphones’ attenuation sufficiently blocked constant ambient noise in clinic rooms, but was not strong enough to block all transient sounds nor constant noise in library rooms. It should be noted that different headphones have different attenuation levels, and hence attenuation should be taken into consideration when choosing headphones.

Blocking acoustic energy in the mid-low frequency spectrum, which coincides with the vast majority of violations, is harder compared to higher frequencies when using passive attenuation. Active noise attenuation systems on the other hand are very effective in attenuating acoustic energy at the low frequency spectrum [25]. These systems include a microphone that picks up ambient noise, a phase inverter, and a speaker which reproduces the ambient noise inverted, resulting in noise cancellation. This technique has applications in several areas where low frequency acoustic energy is present [26], while research on active noise attenuation in audiometry has shown promising results [9,18,22,27] (2.4. ‘Active Noise Reduction (ANR) Earmuffs’).

### 4.5. Addressing Ambient Noise—Online Test Pause

The most frequent case of violations were transient sounds. Irritation from such sounds can only be avoided by reducing the sound level itself, such as increasing headphones attenuation or placing carpets outside the testing room. Signs notifying people to be silent when passing by the testing room and keeping windows closed to avoid traffic noise would also help. Masking effects can be easily avoided by simply pausing the test or ignoring responses when needed. When such a system should be activated, and what should be expected in terms of effectiveness, can be informed by research on the questions raised in a previous section in this discussion (see ‘Effects of ambient noise on audiometry’). 

Ambient noise monitoring systems embedded in automated audiometric systems are already available in industry that ‘ensure testing compliance’ [28,29]. Unfortunately, no details are available on what exactly the decisions being made by the software are, nor which criteria should be met for these decisions to be made. The authors of this paper think that the ambient noise spectrum, time of transient sound occurrence in relevance to time stimuli presentation, and stimuli frequency might serve as good candidates for such a system. This might be a theme for future research in the area of user-operated audiometry.

## 5. Conclusions

Our results suggest that large-scale implementation of user-operated audiometry outside the traditional sound-treated audiometric rooms is possible, at least under some realizable conditions. So far, it is not possible to be conclusive on what exactly these conditions are, as more research on the effects of ambient noise on audiometry is needed. It is likely that necessary conditions differ between rooms, and that they are easier to meet in (non-sound-treated) rooms in audiological clinics compared to rooms in libraries. Attenuation provided by circumaural headphones is likely to be one of the necessary conditions for all cases. Beyond this, and depending on the room under discussion, an online system that makes decisions based on ambient noise might also be included, perhaps in combination with active attenuation.

## Figures and Tables

**Figure 1 healthcare-11-00889-f001:**
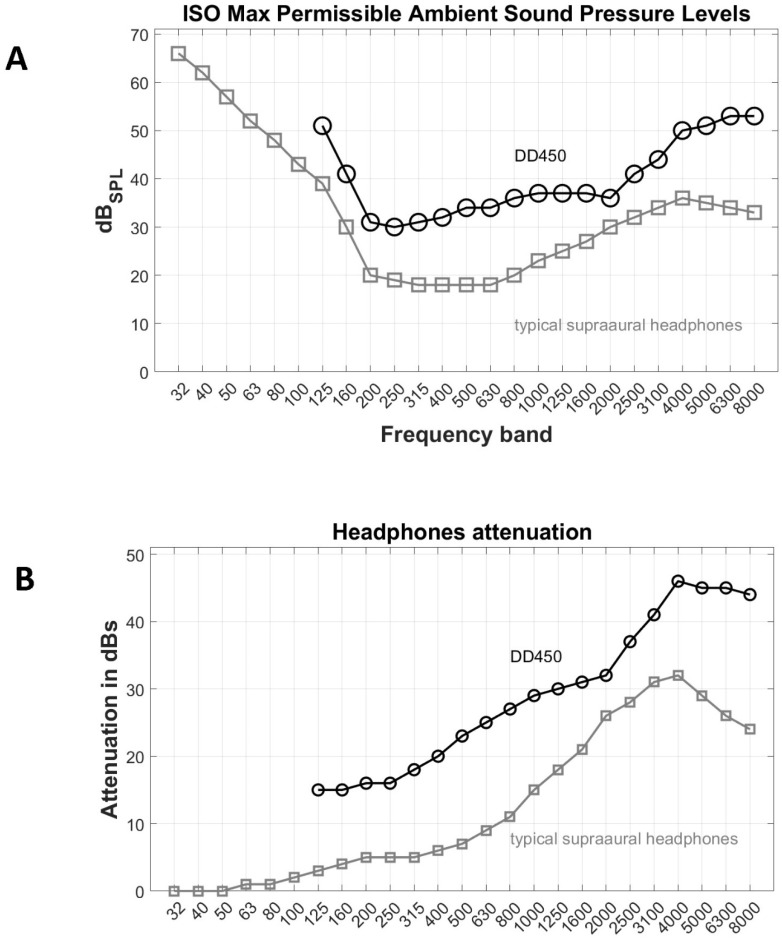
(**A**) Max Permissible Ambient Sound Pressure Level according to ISO 8253-1 [13] when DD450 or typical supra-aural headphones are used. (**B**) DD450 and typical supra-aural headphones’ attenuation.

**Figure 2 healthcare-11-00889-f002:**
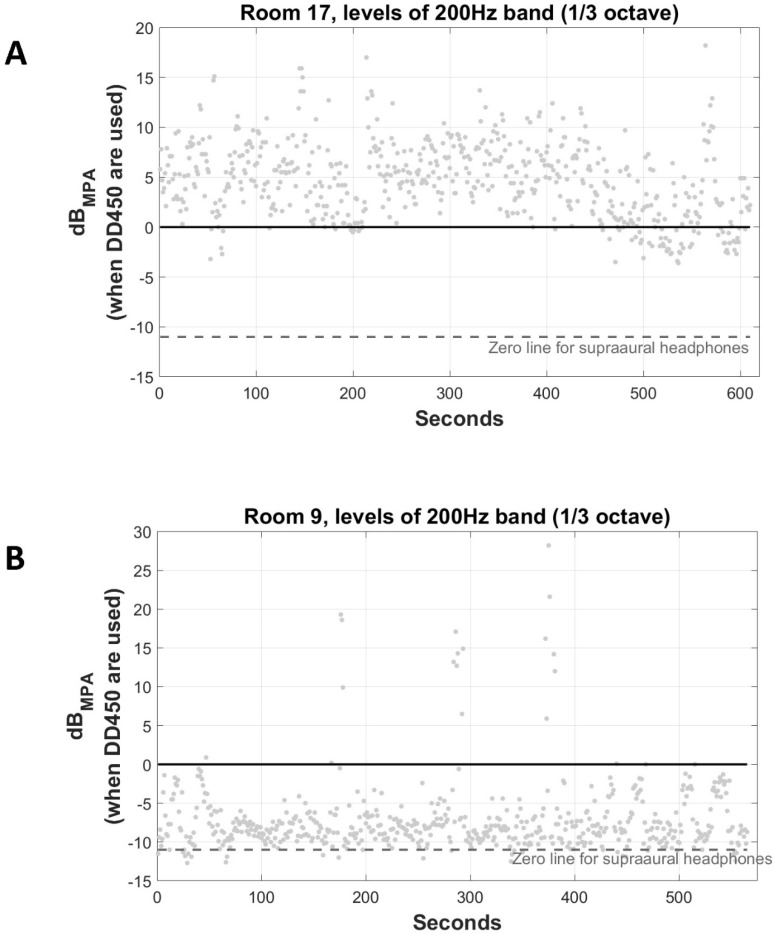
(**A**) Ambient noise of 200 Hz band in a room with quasi-constant violations. (**B**) Ambient noise of 200 Hz band in a room without quasi-constant violations. dB_MPA_ refers to levels with respect to Maximum Permissible Ambient noise levels when DD450 circumaural headphones are used. Dashed grey line refers to Maximum Permissible Ambient noise levels when typical supra-aural headphones are used.

**Figure 3 healthcare-11-00889-f003:**
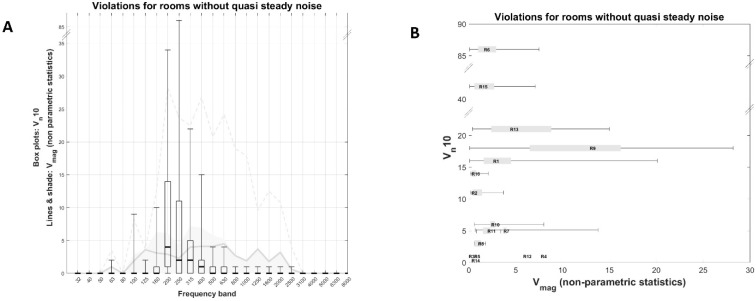
(**A**) Descriptive statistics of V_n_10 in boxplots and V_mag_ in lines and shaded areas for all rooms without quasi-constant violations combined (rooms 1 to 16). Solid grey line: median, shaded area: 1st to 3rd quartile, dashed line: max. (**B**) V_mag_ non-parametric descriptive statistics vs. V_n_10 of each room’s most frequently violated frequency band. No violations were recorded in Room 14. V_n_10 is the number of data points (1 sample per second) that violate Maximum Permissible Ambient noise levels. V_mag_ is the magnitude of the violation in dB.

**Figure 4 healthcare-11-00889-f004:**
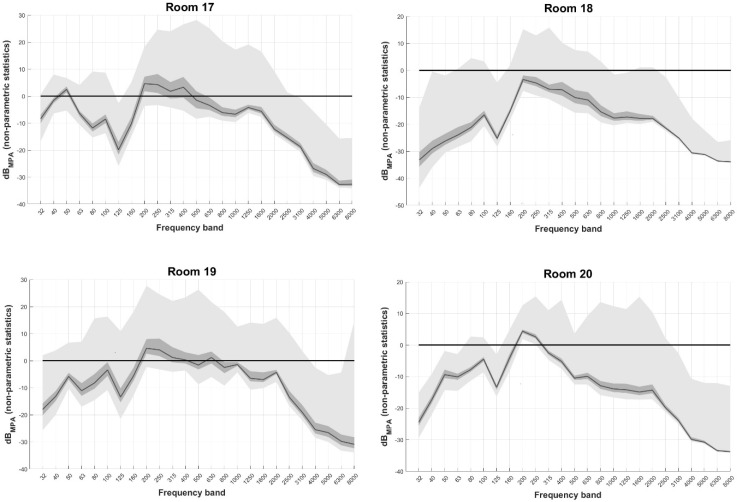
dB_MPA_ non-parametric descriptive statistics per frequency band when supra-aural headphones (DD450) are used for rooms with quasi-constant violations (rooms 17 to 20). Black line: median, dark grey area: 1st to 3rd quartile, light grey areas: min to 1st quartile and 3rd quartile to max. dB_MPA_ refers to levels with respect to Maximum Permissible Ambient noise levels when DD450 circumaural headphones are used.

**Figure 5 healthcare-11-00889-f005:**
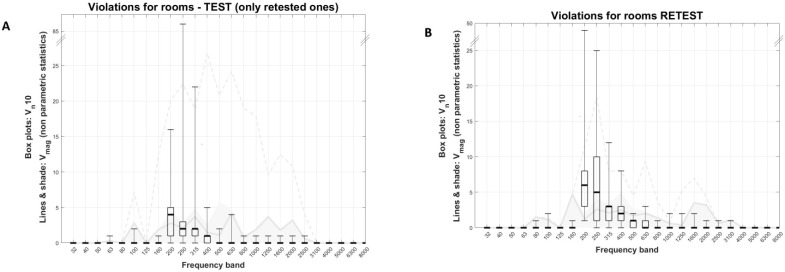
Test (**A**) and retest (**B**) descriptive statistics of V_n_10 in boxplots and V_mag_ in lines and shaded areas in 9 rooms (1, 2, 4–7 and 10–12). Solid grey line: median, shaded area: 1st to 3rd quartile, dashed line: max. V_n_10 is the number of data points (1 sample per second) that violate Maximum Permissible Ambient noise levels. V_mag_ is the magnitude of the violation in dB.

**Figure 6 healthcare-11-00889-f006:**
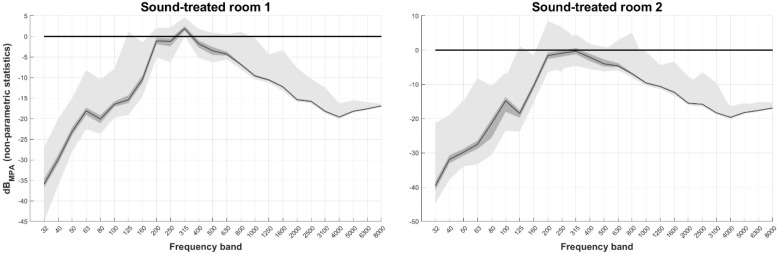
dB_MPA_ non-parametric descriptive statistics of dB_MPA_ per frequency band when typical supra-aural headphones are used in 2 traditional audiometric sound-treated rooms. Black line: median, dark grey area: 1st to 3rd quartile, light grey areas: min to 1st quartile and 3rd quartile to max. dB_MPA_ refers to levels with respect to Maximum Permissible Ambient noise levels when typical supra-aural headphones are used.

**Table 1 healthcare-11-00889-t001:** Room demographic data and presence or absence of quasi-constant violations.

Location	Room Number	Floor	Area (Approximately)	Retest	Quasi-Constant Violations
Audiology clinic, OUH	1–13	3rd	~5–10 m^2^	Yes, 9 rooms	Absent
Private ENT clinic	14 and 15	ground	~5–10 m^2^	No	Absent
SDU hospital	16	ground	~5–10 m^2^	No	Absent
SDU library	17	1st	>50 m^2^	No No	Present
	18	2nd			
Odense public library	19	1st	>50 m^2^	No No	Present
	20	2nd			
**TOTAL**	**20**			**9**	**Absent in 16**
					**Present in 4**

**Table 2 healthcare-11-00889-t002:** Most violated frequency bands for each room without quasi-constant violations (rooms 1 to 16).

Most Frequently Violated Frequency Band	Rooms	N of Rooms
200 Hz	1, 3, 5, 9, 10, 11, 12, 13	9
250 Hz	2, 4, 6, 15	4
315 Hz	7, 8	2
No violations	14	1
**TOTAL**		**16**

**Table 3 healthcare-11-00889-t003:** Ambient noise test–retest V_n_10 and median (and max in parenthesis) V_mag_ values in 9 rooms for the most frequently violated frequency in either test or retest measurements.

Room	Most Frequently Violated Frequency Band	V_n_10			Median (max) V_mag_		
		TEST	RETEST	Dif	TEST	RETEST	Dif
Room 1	200 Hz	16	1	15	2.9 (20.1)	0.4 (0.4)	2.5 (19.7)
Room 2	250 Hz	11	25	14	0.6 (3.7)	2.2 (18.4)	1.6 (14.7)
Room 4	200 Hz	0	49	49	0 (0)	1 (3.9)	1 (3.9)
Room 5	200 Hz	1	5	4	0.7 (0.7)	3.1 (4.8)	2.4 (4.1)
Room 6	250 Hz	86	12	74	1.9 (7.5)	2.8 (10.7)	0.9 (3.2)
Room 7	200 Hz	4	15	11	6.5 (8.5)	1 (3.2)	5.5 (5.3)
Room 10	250 Hz	2	21	19	4.3 (6.9)	2.2 (9.6)	2.1 (2.7)
Room 11	200 Hz	5	3	2	2.3 (3.4)	0.5 (1.7)	1.8 (1.7)
Room 12	250 Hz	1	3	2	5.6 (5.6)	3.9 (5.4)	1.7 (0.2)

## Data Availability

Data are available upon request from the first author.

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
