# Peer review of "Ambient Noise in Candidate Rooms for User-Operated Audiometry"

_healthcare, 2023, doi:10.3390/healthcare11060889_

Round 1

Reviewer 1 Report

MS is easy to read, introduction, materials, results, discussion and conclusions are understandable and clear. Only weaknesses were references, there was variability in the type of references.

I would have liked to see some discussion on your results to basic audiometry at octave band 250 Hz, and on the other side on other octave band frequencies.

I was also wondering the types of impulses like doors, steps, what else? You mentioned in one sentence possible control methods against disturbing noise like shoes or carpets. Could you elaborate it little further?

Could there be better guidelines etc based in your study in the contemporary rooms used to audiometry?

Author Response

Thank you very much for your all comments.

In regards to your 250Hz comment, could you please provide some clarifications? Is it about the effect of ambient noise on particular frequencies (to our knowledge literature on ambient noise effects on audiometry are extremely limited)?

Regarding the type of brief sound sources the following was added in the discussion l. 256-259:

‘This study was not designed to investigate on sound sources, though some speculations can be made. As clinics was quite busy and people were walking though the corridor, it is most probable that most sounds were people speaking or sounds related to movement, e.g., steps. Some sounds might also have come from outside, such as traffic.’

Regarding the methods against disturbing noise the following is now added in l. 391-393:

‘Signs notifying people to be silent when passing by the testing room and keeping windows closed to avoid traffic noise would also help.’

Regarding the guidelines comment, it is not easy to apply our results on traditional audiometric rooms, as they are limited to when using a specific circumaural headset.

Reviewer 2 Report

Explain better what the authors intend to do.

Better describe the measurement environment, missing photos, drawings.

Explain better how you made the measurements.

Explain how the boxes where the measurements are made are made (what kind of materials).

What floor are these rooms on, are they isolated from the outside?

Explain why the problem occurs at 200 Hz.

Explain what kind of external noise you are measuring.

The measurement of ambient noise is performed with an iPad, is it a correct system?

How did you calibrate it? are you sure it gives you a correct acoustic measurement?

Author Response

Thank you very much for your all comments.

The following additions have been done in the manuscript:

Regarding the type of brief sound sources the following was added in the discussion l. 256-259:

‘This study was not designed to investigate on sound sources, though some speculations can be made. As clinics was quite busy and people were walking though the corridor, it is most probable that most sounds were people speaking or sounds related to movement, e.g., steps. Some sounds might also have come from outside, such as traffic.’

Regarding the methods against disturbing noise the following is now added in l. 391-393:

‘Signs notifying people to be silent when passing by the testing room and keeping windows closed to avoid traffic noise would also help.’

Regarding the guidelines comment, it is not easy to apply our results on traditional audiometric rooms, as they are limited to when using a specific circumaural headset.

Regarding the rooms, floor is reported at table 1. The following is now added to l. 85-86:

‘(see Table 1 for details on demographic data).’

Regarding whether rooms were isolated from outside, the following was added at l. 86-87:

‘All rooms had one or two external walls, hence sound from outside might contribute to ambient noise inside the room.’

Regarding the calibration, the following was added at l. 98-105:

‘A 6th generation iPad (Apple), model number A1893 was used for measuring background noise. The software used is the AudioTools app (Studio Six Digital) with the additional SPL Graph module (newest release as of 02/12/2021). The iPad was calibrated by adjusting the sensitivity of the iPad microphone according to the measured sound level of white noise of a B&K 2236 Sound-level meter. Calibration was performed in an acoustic well-regulated office room, where a homogenous sound-field in the measuring area can be created with no difference in the sound pressure at the two microphone positions (B&K and iPad).’

Also, changes on language have been done in the manuscript.

Best regards

Reviewer 3 Report

In the article, the Authors searched for a solution to the limitations related to the lack of accessibility for people with hearing problems to offices diagnosing the degree of hearing loss. Therefore, they researched whether ordinary rooms, without the required soundproofing, can be used for this type of diagnostics. The work seems to be a complete elaboration of the adopted topic. Its composition is correct, and the selection of literature is appropriate. Some doubts were raised by using a phone model with software in the acoustic measurements, but further observations, measurements, and analysis seem logical and consistent. The obtained results, and especially the conclusion confirming the possibility of using such rooms as "hearing diagnostic rooms", is a good premise for potential patients needing this type of test. I hope that the Authors will implement the project and it will be possible to use this type of observation. Please check the English language, there are minor linguistic and stylistic errors, but they do not significantly affect the reception of the content.

Author Response

Thank you very much for your all comments.

The manuscript has now been checked for language errors. Details on calibration was added, please see l. 98-105.

Round 2

Reviewer 2 Report

improves the quality of the figures.

If you can you should remove the comments below the figures and insert them in the corresponding paragraphs.

Author Response

The following was added in l. 142-144:

‘Figure 1 shows the respective MPASPL values when DD450 and typical supraaural headphones are used respectively, and the attenuation they provide.’

The following was added in l. 161-163:

‘Figures 2A and 2B show ambient noise of 200 Hz band in a room with quasi-constant violations and a room without quasi-constant violations respectively.’

For the rest of the figures, comments are already in-text.